# Histological Change in Soft Tissue Surrounding Titanium Plates after Jaw Surgery

**DOI:** 10.3390/ma12193205

**Published:** 2019-09-30

**Authors:** Gabriel Armencea, Dan Gheban, Florin Onisor, Ileana Mitre, Avram Manea, Veronica Trombitas, Madalina Lazar, Grigore Baciut, Mihaela Baciut, Simion Bran

**Affiliations:** 1Department of Oral and Maxillofacial Surgery, Iuliu Hatieganu University of Medicine and Pharmacy, 400000 Cluj-Napoca, Romania; garmencea@gmail.com (G.A.); gbaciut@umfcluj.ro (G.B.); 2Department of Pathology, Iuliu Hatieganu University of Medicine and Pharmacy, 400000 Cluj-Napoca, Romania; dgheban@gmail.com; 3Department of Implantology and Maxillofacial Surgery, Iuliu Hatieganu University of Medicine and Pharmacy, 400000 Cluj-Napoca, Romania; ilmitre@yahoo.com (I.M.); avram.manea@umfcluj.ro (A.M.); madilazar@yahoo.com (M.L.); mbaciut@yahoo.com (M.B.); dr_brans@yahoo.com (S.B.); 4Department of ENT Surgery, “Iuliu Hatieganu” University of Medicine and Pharmacy, 400000 Cluj-Napoca, Romania; veronicatrombitas@gmail.com

**Keywords:** titanium, miniplates, osteosynthesis, Ti particles, implants

## Abstract

The aim of this study was to evaluate the microscopic structure of soft tissue covering titanium plates and screws used in jaw surgery (mandible fracture and orthognathic surgery), after a minimum period of 12 months from insertion, and to quantify the presence of any metallic particles. Periosteum covering the osteosynthesis plates was removed from 20 patients and examined by light microscopy in order to assess the cell morphological changes and the possibility of metal particles presence in the soft tissue. Local signs of tissue toxicity or inflammation were taken into consideration when evaluating the routine removal of titanium maxillofacial miniplates. No signs of screw loosening or acute inflammation were detected on the osteosynthesis site, but de-coloration of the periosteum was seen, and metallic particles were observed to have migrated into the soft tissues. Even if the titanium is well-tolerated by the human body in time, without severe local or general complications, our findings suggest that plate removal should be considered after bone healing has occurred.

## 1. Introduction

Commercially pure titanium (CpTi) and its alloys are the first-choice material for bone internal fixation in trauma or orthognathic surgery due to their biological and mechanical properties, including their excellent biocompatibility, corrosion resistance, and high mechanical strength [1]. Even if they are well-tolerated, the long-term side effects within human tissues are still unclear and controversial [2]. Plate removal is another debatable issue that has different approaches. Some approaches support immediate removal after bone healing [3], whilst other studies have concluded that titanium and Ti-6Al-4V may be retained as permanent implants in maxillofacial fixation [4,5,6].

The use of CpTi miniplates and screws (Ti-6Al-4V) in maxillofacial pathology helps avoid long-term intermaxillary immobilization and dental-retained splints, but, despite their undoubted advantages, their negative impact on the human body is still debated [6,7,8]. Some papers have presented discolored gray tissues over the miniplates [9,10], and others have reported Ti toxicity due to oxidative stress [11] and the occurrence of redox imbalance, as well as oxidative damage in the periosteum surrounding the Ti-6Al-4V alloy [7]. 

Recent studies have reported chronic inflammation around titanium bone fixation devices due to the increased production of oxygen-free radicals and reactive nitrogen species [9,12]. Oxygen-free radicals can impair normal cell functioning as a result of the increased synthesis of proinflammatory mediators, which can affect cell growth, differentiation, and apoptosis processes [13]. Therefore, the aim of our work was to evaluate the microscopic structure of soft tissue covering CpTi plates.

## 2. Materials and Methods

The experimental group in this study comprised 20 patients that had jaw surgery for a fracture (12 patients) or orthognathic treatment (8 patients), and underwent plate removal after one year from surgery. There were 13 male patients and 7 female patients, with an age range of 17–64 yrs. A total of 70 CpTi miniplates with 2.0 diameter screws made by Stryker Leibinger (GmbH & Co.KG Freiburg, Germany) were removed, and 20 samples of soft tissue overlying the titanium plates were harvested and analyzed by optic microscopy. Ethical approval was granted by the relevant Local Ethics Research Committee, and an informed consent form was signed by all the patients prior to surgery. 

All the specimens excised were immediately fixed in 10% formaldehyde, embedded in paraffin wax, sectioned with a microtome, and stained with Mayer hematoxiline eozine.

Mayer hematoxiline eozine staining [14].

Acidulated Mayer hemalaun substance: 75 g of alaun was dissolved in heat in 1000 mL of distilled water. Additionally, 75 mL of hematoxyline was filtrated in heat (from the base solution of 10%) and 0.30 g of potassium iodate in water. Then, it was all acidulated in 100 mL of solution, with 0.5 mL of acetic acid. 

Eosin was boiled in a jar containing the following:100 mL of alcohol 70°;1 g of alcoholic eosine (Eosinblau siritus soluble);1 g of watery eosine (Eosingelb aqua soluble);0.25 g of orange G;1 mL of officinalis HCl acid.

After cooling 20 mL of saturated liquid, lithium carbonate was added.

Lithium carbonate substance: 100 mL of distilled water + 15 g of lithium carbonate. 

The technique was as follows:-Mayer hemalaun stain, 10 min;-Wash in water;-Clorhydric alcohol differentiation, 1–2 s;-Wash in water;-Change in lithium carbonate, 2–4 min;-Wash in water;-Eozine coloration, 2–3 min;-Wash in water;-Ethanol dropped on slide until no more eozine;-Dehydration with absolute and 95° alcohol;-Clarification in fenicated xylene, xylene;-Mounting on Canadian fabric.

Results:Nucleus: dark blue;Cytoplasm: pink-red;Red cells: bright red;Collagen: pale pink;Elastic fibers: bright pink.

Morphometric analysis

The samples were studied under light microscopy with a Leica DM750 microscope. In order to measure the dimensions of the metallic particles present in the soft tissue, ImageJ software used (https://imagej.nih.gov/ij/download.html). Images of the samples under 400× magnification were used.

## 3. Results

In all cases, it was noted that the miniplates were macroscopically covered by dense fibrotic tissue, and had no inflammatory signs. Visible dark-grey pigmentation was seen in 16 out of 20 samples taken. 

Under light microscopy, varying degrees of fibrosis were seen, the samples displayed connective tissue without an inflammatory reaction of any kind, and pigmented debris were found in all samples, comprised of large or dust-like particles. 

To produce an accurate measurement of these particles, we used the ”Straight” function in ImageJ software, which measures the distance between two points, and employing “CTR+M”, this value was passed into an Excel document, as seen in Figure 1.

All the particles found were extracellular.

Three measurements were taken in all cases for a better accuracy, and the average value was calculated. The resulting number of these measurements represents the total number of pixels in the image, and for micrometrical conversion, we used an image photographed in identical conditions of a micrometric grid (Figure 2).

The particles found are of different sizes, varying from 54.5 Micron (Figure 3) to dust-like 1 Micron particles (Figure 4).

The distribution pattern of these tiny dust-like particles suggests the disintegration of larger particles. All the particles found were extracellular.

## 4. Discussion

Our study shows that there are dark-colored particles in the soft tissues after at least one year post-insertion, which vary in their dimensions between 1 micron and 54.5 microns. It is debatable whether these particles are detached when the Ti material is inserted, whether they are the result of a metabolic process of fagocytosis [2], or whether they are caused by the Ti oxides at the plate surface [3,5]. The miniplates used in cranio-facial surgery are highly resistant to corrosion and mechanical wear [15,16]; some studies suggest that the titanium release into the surrounding tissue is caused by plate damage through manipulation or movement while “in situ” [3,16]. Our patients did not present any screw loosening or plate mobility at removal. It is believed that if wear damages the surface TiO_2_ layer, the surrounding tissue turns black [17].

All the particles found were extracellular, but other studies have proved that they can be intracellular [2,18], and some small extracellular particles could be the result of phagocytic cell apoptosis, which could explain the powder-like particles present in our study. Other paper presents the fact that the transport of titanium particles can occur via lymphatics to regional lymph nodes [19], which would sustain the mandatory plate removal protocol postulated by Kim et al. [3], who found tissue degeneration around the titanium particles. Our study found various degrees of fibrosis next to the implant site, without signs of inflammation or tissue degeneration. Titanium particles present at a distance from the implant site were found by Bessho et al.—lungs, spleen, kidneys, and liver [20]. Onodera et al. reported a submandibular lymph node with titanium particles 2 years after a mandibular titanium plate implant in a 41-year-old man [21].

Other papers have described that large implant-derived particles are produced exclusively by the mechanical wear process in the articular coupling of prostheses and are not observed with osteosynthesis implants [22]. In contrast with this, our findings show the presence of Ti particles at the osteosynthesis site. A possible explanation for this could be related to the mechanical “damage” of the plate-screw system at the insertion time, but the particles were also found on top of the plates, and not only on the screws. 

Adverse reactions to titanium have been seen over time through titanium hypersensitivity reactions, pacemaker case reports, dental implant case reports, orthopedic implant case reports, and neurosurgical implants [23]. A case of granulomatous pulmonary disease with the associated pulmonary deposition of titanium was reported in a 45-year-old man who worked as a furnace feeder for an aluminum smelting company [22]. All these adverse reactions, even if few in number, prove that Ti implants can cause local and general problems.

It is believed that the passive layer of TiO_2_ effectively protects the surface of the implant from the effects of corrosive agents present in the environment of body fluids. The TiO2 layer may be damaged, resulting in the release of metal ions (such as titanium) into the surrounding tissues. Mandibular movements can cause micromovements of the fixed bone fragments, which increases the friction between the miniplates and the screws, and this phenomenon (tribocorrosion) could be responsible for the corrosion of titanium fixations and the release of titanium ions into the tissues/organs surrounding the implants, promoting inflammatory reactions [7,24].

Metal debris may be released into the tissues as a result of manufacturing defects, corrosion, surface contamination, or mechanical damage which may occur during insertion or removal, or may be due to wear whilst in situ [5,25]. Tissue trauma, hematoma, tissue healing, and mechanical instability between the implant and the surrounding tissues may be at least partly responsible for the histological changes seen in the tissues [1].

It is generally suggested that the particles detached from titanium plates are tolerated relatively well, even at high concentrations. However, we must consider the long-term tissue reaction to debris from titanium. The concentration of metals that could cause adverse effects is not currently known, and the clinical significance and long-term effects of titanium particles in the tissues, both locally and systemically, are unknown. 

## 5. Conclusions

The microscopic evaluation of the soft tissues covering the cpTi plates does not present any kind of inflammatory signs, but presents numerous dark-colored particles, of various dimensions.

Tissue irritation, hardware loosening, and other adverse reactions can occur over time. Therefore, future studies are needed in order to make a proper assessment of the soft tissue changes on CpTi plates.

## Figures and Tables

**Figure 1 materials-12-03205-f001:**
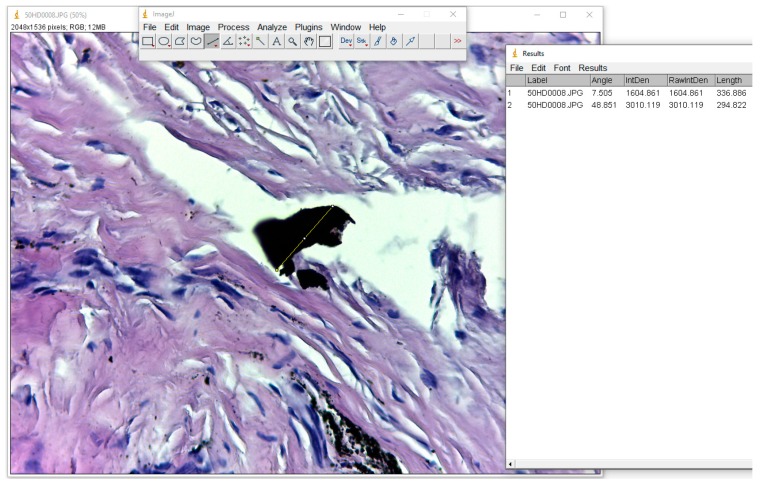
Morphometric analysis of debris particles in ImageJ software.

**Figure 2 materials-12-03205-f002:**
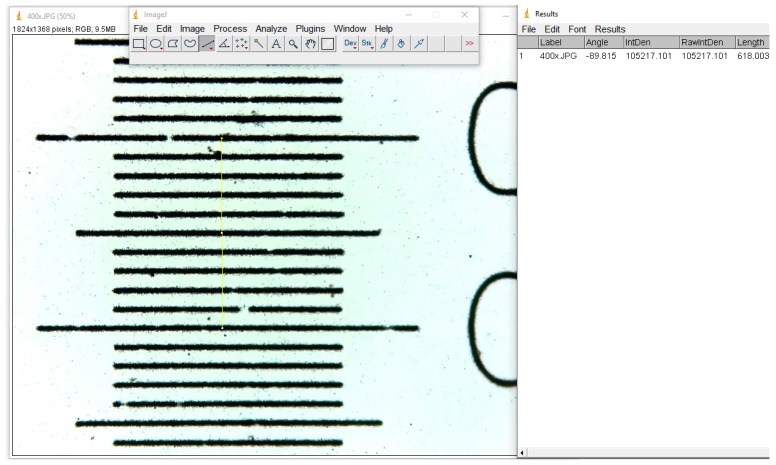
Measurement calibration in ImageJ with a micrometric grid.

**Figure 3 materials-12-03205-f003:**
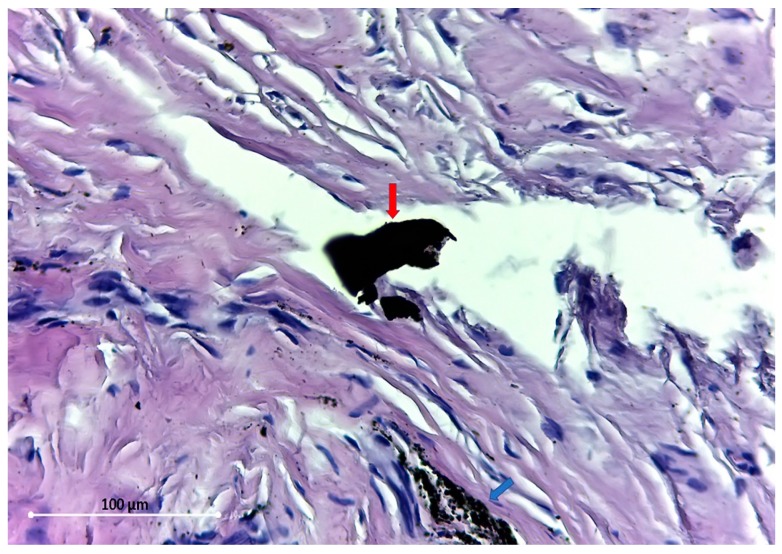
Large debris (red arrow), near a cluster of small particles (blue arrow). HE stain ×400.

**Figure 4 materials-12-03205-f004:**
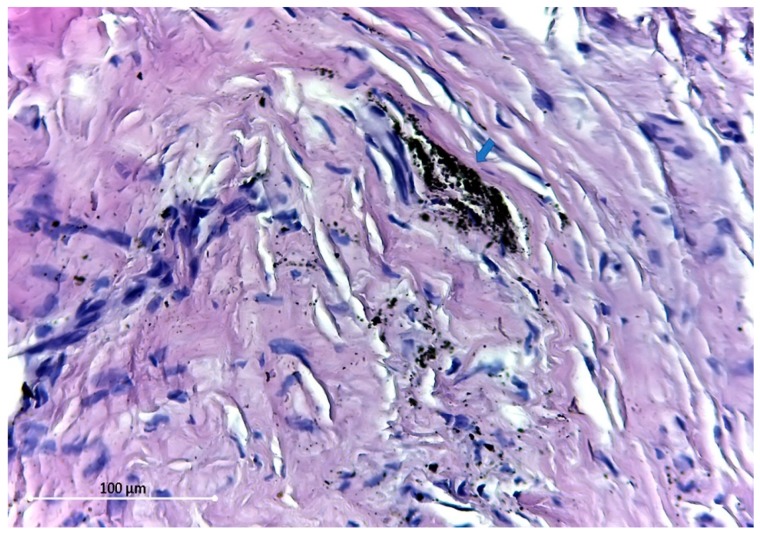
Dust-like (1 micron) particles—blue arrow. HE stain ×400.

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
