# Peer review of "Histological Change in Soft Tissue Surrounding Titanium Plates after Jaw Surgery"

_materials, 2019, doi:10.3390/ma12193205_

Round 1

Reviewer 1 Report

Dear authors, nice work. Here you can find the attached file with my suggestions. You should ammend it before it can be considered ready for publication.

Author Response

Dear Reviewers, thank you for the prompt and competent review, we have made the appropriate changes in the manuscript

Reviewer 1 response:

Line 34: we have added “and”. Line 44: we have deleted “titanium”. Line 48: deleted “and”, added “which”. Line 49: “cpTi” added. Line 55: “cpTi” added and “titanium” deleted. Line 164: “Tribocorrosion” added. Line 179: “we believe that titanium miniplates should be removed routinely.” replaced with  ”future studies are needed in order to make a proper assessment of the soft tissue changes on cpTi plates.”

Reviewer 2 Report

This is a histological study that investigating the soft tissue changes near the Ti alloy plates. It included 20 patients and documented two types of Ti-alloy particles within soft tissue. It has its merits to be published. 

However, the title is confusing.  The term of " soft tissue morphology" usually refers to facial appearance. Why not use the the " histological change in soft tissue" or "cell morphology" directly?

Please improve the clarity of the figures and their captions. Please include the scale in each figure. Please indicate the tissues with a specific color. Also, please rearrange the components in each figure to reduce the blank portion.

Please improve the clarity of the reasoning. 

Specially, 

L22: One paragraph is OK.

L29: Need more keywords

L44: be consistent, using Ti-6Al-4V

L49-50: please be consistent with abstract L17-20.

L55: provide details of the vendor.

L61: please provide reference here

L95-96: font size error

L110-112: What is this?

L113: how about "taken" instead of "done"?

L129-130: Have your study show "there is a release of metallic particles into the soft tissues"? No. What you have are two types of  the dark regions as shown in the figures. Are these dark regions represents the Ti alloy?  Maybe. Please provide further evidence and/or reasoning.  Does these two types of dark region SUGGESTS "a release of metallic particles"? Maybe. Please provide further evidence and/or reasoning.   The further discussion in this paragraph are very good but the first sentence is a piece of overstatement.

L138: "All the particles found were present extracellular" ? please provide description in the Results section.

L167-171: Does these factors /situation apply to  the situation in your study? What is your message here?

L173-177: These sentences belongs to Discussion. What is your conclusion?

L178-179:  This statement is true in general but how does this relate to your study and your conclusion?

Author Response

Dear Reviewers, thank you for the prompt and competent review, we have made the appropriate changes in the manuscript

Reviewer 2 response:

Title changed in “Histological change in soft tissue surrounding titanium plates after jaw surgery”. We have improved the clarity of the figures , included the scale in each figure and the blank portion of the figures was reduced. L22: text transformed to one paragraph. L29: added “Ti particles; implants”. L44: Ti-6Al-4V used instead of Ti6Al4V. L49-50: Text changed to “aim of our work was to evaluate the microscopic structure of soft tissue covering cpTi plates”. L55: Details of the vendor added: Stryker Leibinger,GmbH & Co.KG Freiburg, Germany. L61: Reference provided: 22 L95-96: font size changed. L110-112: lines deleted. L113: "taken" instead of "done". L129-130: text changed to “dark - colored particles” instead of “release of metallic particles”

This pigmented particles were interpreted as metallic debris based on personal experience of autors

L138: The text: “All the particles found were present extracellular” was inserted at L112 L167-171: These factors /situation could be one of the reason the metallic particles appear in the soft tissue, but further studies are needed in order to make a proper assessment of the soft tissue changes – L179 L173-177 and 178-179: The paragraph: “It is generally suggested that the particles detached from titanium plates are tolerated relatively well, even at high concentrations. However, we must consider the long-term tissue reaction to debris from titanium. The concentration of metals that could cause adverse effects is not known so far, and the clinical significance and long-term effects both locally and systemically of titanium particles in the tissues are unknown” was moved to Discussion and the Conclusions were rephrased L179-180.

Round 2

Reviewer 2 Report

Revisons have been made and the revised version is better.  However,  the figures can be further improved.

L113: Has the average of three measurements been reported? Please make it clear.

L178-180 Conclusions: Please focus on this study. The  aim of this work was to evaluate the microscopic structure of soft tissue covering cpTi plates, as stated in L49. Please draw conclusion accordingly. After that, you may provide its clinical implication.

Author Response

Dear Reviewers, thank you for the prompt and competent review, we have made the appropriate changes in the manuscript:

Editor decision response:

We have modified the conclisions L 198-199, We have inserted 3 citations related to titanium plates: 23-24-25, The minor grammar modifications revealed by reviewer 1 have been made

Reviewer 2 response:

L 120: ”and the average value was calculated” inserted in text. We have modified the conclisions L 198-199,